# Semi-Quantitative [^18^F]FDG-PET/CT ROC-Analysis-Based Cut-Offs for Aortitis Definition in Giant Cell Arteritis

**DOI:** 10.3390/ijms232415528

**Published:** 2022-12-08

**Authors:** Olivier Espitia, Jérémy Schanus, Christian Agard, Françoise Kraeber-Bodéré, Alexis F. Guédon, Antoine Bénichou, Jean-Michel Serfaty, Sandrine Coudol, Matilde Karakachoff, Bastien Jamet

**Affiliations:** 1Department of Internal and Vascular Medicine, CHU Nantes, Nantes Université, F-44000 Nantes, France; 2Department of Nuclear Medicine, CHU Nantes, Nantes Université, CNRS, INSERM, CRCINA, F-44000 Nantes, France; 3Department of Cardiovascular Imaging, CHU Nantes, Nantes Université, F-44000 Nantes, France; 4Pôle Hospitalo-Universitaire 11, Santé Publique, Clinique des Données, CHU Nantes, Nantes Université, INSERM, CIC 1413, F-44000 Nantes, France

**Keywords:** [^18^F]FDG-PET/CT, aortitis, giant cell arteritis, aortic atheroma, large vessel vasculitis, diagnostic semi-quantitative thresholds

## Abstract

[^18^F]fluorodeoxyglucose-positron emission tomography/computed tomography ([^18^F]FDG-PET/CT) is used to diagnose large vessel vasculitis in giant cell arteritis (GCA). We aimed to define a semi-quantitative threshold for identifying GCA aortitis from aortic atheroma or the control. Contrast enhanced computed tomography (CECT) was used as the reference imaging for aortic evaluation and to define aortitis, aortic atheroma and control aortas. [^18^F]FDG-PET/CT was performed on 35 GCA patients and in two different control groups (aortic atheroma (*n* = 70) and normal control (*n* = 35)). Aortic semi-quantitative features were compared between the three groups. GCA patients without aortitis on CECT were excluded. Of the GCA patients, 19 (54.3%) were not on glucocorticoids (GC) prior to [^18^F]FDG-PET/CT. The SUV_max_, TBR_blood_ and TBR_liver_ aortic values were significantly higher in the GCA aortitis group than in the aortic atheroma and control groups (*p* < 0.001). Receiver operating characteristic curve analyses brought to light quantitative cut-off values allowing GCA aortitis diagnosis with optimal sensitivity and specificity versus control or aortic atheroma patients for each PET-based feature analyzed. Considering the overall aorta, a SUV_max_ threshold of 3.25 and a TBR_blood_ threshold of 1.75 had a specificity of 83% and 75%, respectively, a sensitivity of 81% and 81%, respectively, and the area under the ROC curve (AUC) was 0.86 and 0.83, respectively, for aortitis detection compared to control groups in GCA cases with GC. A SUV_max_ threshold of 3.45 and a TBR_blood_ threshold of 1.97 had a specificity of 90% and 93%, respectively, a sensitivity of 89% and 89%, respectively, with an AUC of 0.89 and 0.96, respectively, for aortitis detection compared to the control in GC-free GCA cases. Discriminative thresholds of SUV_max_ and TBR_blood_ for the diagnosis of GCA aortitis were established using CECT as the reference imaging.

## 1. Introduction

Giant cell arteritis (GCA) is the most frequent primary vasculitis and affects large arteries. In GCA, two overlapping phenotypes of the vasculitis can be distinguished: cephalic and large vessel (LV) GCA [1]. At GCA diagnosis, aortitis is observed in about half of the cases [2,3,4]; the presence of aortitis at diagnosis is associated with more relapses or more vascular events during the course of the disease [5,6,7]. For LV assessment, the European League against Rheumatism (EULAR) [8] recommends performing contrast enhanced computed tomography (CECT) or [^18^F]fluorodeoxyglucose-positron emission tomography with computed tomography ([^18^F]FDG-PET/CT). Thus, in this population of elderly patients, where atheromatous disease is also commonly present, it is important to properly define aortitis on LV imaging.

To date, visual analysis is recommended for assessing [^18^F]FDG-PET/CT in GCA detection using the systematic predefined liver background uptake as a diagnostic threshold (aspect of vasculitis when vascular wall uptake is superior to liver background uptake) [8,9], which depends on the expertise of the physicians and may lack precision of metrics due to visual FDG uptake categorization with the liver uptake as reference.

Alternatively, semi-quantitative diagnostic analytical approaches are more recent and scarce but seem to bring new light in [^18^F]FDG-PET/CT GCA assessment [10,11,12]. To limit the reader bias, increase reproducibility and optimize precision, semi-quantitative analysis may be preferred. On the PET image, semi-quantitative methods use regions of interest (ROI) to determine maximum standard uptake values (SUV) [10]. In vasculitis, Target-to-Background Ratios (TBR), which is the ratio of the SUV of the arterial wall to the reference tissue (e.g., liver or blood pool), are also used to quantify arterial FDG uptake [13]. Guidelines for FDG-PET imaging recommend the use of TBR, normalizing to venous blood pool instead of SUV, for the quantification of arterial wall FDG uptake [9].

A few studies have already tried to identify quantitative thresholds for the diagnosis of GCA aortitis using [^18^F]FDG-PET/CT, but the different thresholds identified were not discriminating [14,15,16,17]. Moreover, none of these studies have compared [^18^F]FDG-PET/CT with CECT as the reference imaging, with each of them using temporal artery biopsy and clinical diagnosis as the reference method [14,15,16]. Using CECT as a reference is important because only 50% of GCA patients have aortitis, so we cannot analyze aortic [^18^F]FDG uptake of all GCA patients to identify the cut-off for aortitis. In addition, CECT allows for differentiation of aortitis from aortic atheromatous plaques which are also responsible for [^18^F]FDG uptake. Thus, the semi-quantitative method to evaluate GCA patients requires further clarification, in order to find out a semi-quantitative threshold for GCA aortitis detection.

The aim of this study was to identify semi-quantitative cut-offs derived from [^18^F]FDG-PET/CT imaging using receiver operating characteristic (ROC) curves for identifying GCA-associated aortitis, with CECT as the reference imaging.

## 2. Results

### 2.1. Patients

Thirty-five GCA aortitis patients were matched with 70 aortic atheroma cases and 35 control patients without either atheroma or aortitis. The mean age was 68.3 (±8.8), 70.7 (±9.0) and 67.9 (±7.9) years in the GCA aortitis, aortic atheroma, and control group, respectively. In this study of GCA aortitis, as in most studies of GCA, there was a female predominance. In the aortitis group, 19 (54.3%) GCA patients were glucocorticosteroid-free before PET. The median time from initiation of corticosteroid therapy to completion of the PET was 7 days for 16 (45,7%) GCA patients with glucocorticosteroid (GC) treatment started before PET.

In both the aortic atheroma and the control group, patients had a history of neoplasia: 24 (34.3%) and 8 (22.9%) with hematologic malignancies, respectively, 21 (30.0%) and 25 (71.4%) with melanomas, respectively, 15 (21.4%) with pulmonary neoplasia and none in the control group, 5 (10.0%) with gastrointestinal cancers and none in control group and 5 (4.3%) and 2 (5.7%) with other neoplasia (breast cancer, ENT, skin squamous cell carcinoma), respectively. None of the included patients had active cancer or were undergoing chemotherapy. The characteristics of the patients are summarized in Table 1.

### 2.2. [^18^F]FDG-PET/CT Semi-Quantitative Features 

The aortic, liver and blood SUV_max_ are presented for each group in Table 2 with an intra-aortitis group comparison with and without GC treatment.

Table 3 shows SUV_max_, TBR_liver_ and TBR_blood_ according to aortic FDG uptake visual grading.

In the overall aortic analysis as well as for each of the five aortic segments, all the SUV_max,_ TBR_blood_ and TBR_liver_ values were significantly higher in the GCA aortitis group than in the aortic atheroma and control groups (Figure 1 and Appendix A).

### 2.3. Diagnostic Semi-Quantitative Cut-Off Values from ROC Curves

Analysis has shown that the following parameters, SUV_max_ and TBR_blood_, provide the best discrimination between arteritis, atheroma and controls and these will be presented in detail. With regard to the ROC curves, comparisons of the different PET values between aortitis versus aortic atheroma and aortitis versus control found excellent AUCs for each [^18^F]FDG-PET/CT-derived semi-quantitative feature analyzed (SUV_max_, TBR_blood_ and TBR_liver_, Appendix A). For all of them, clear diagnostic cut-off values were defined with excellent specificities and sensitivities (Table 4, Figure 2 and Appendix A).

Specific thresholds were identified for each of the aortic segments by comparing patients with CGA aortitis without GC versus normal controls or atheromatous patients (Appendix A). When comparing patients with GCA aortitis who had GC before the PET with normal controls and atheromatous patients, in an overall aortic analysis, the AUC was 0.86 and a threshold SUV_max_ of 3.25 had a sensitivity of 81% and a specificity of 83% compared with all control patients. For GCA-aortitis without GC before PET vs. all controls, the AUC was 0.89 and a threshold SUV_max_ of 3.45 had a sensitivity of 89% and a specificity of 90% (Table 4).

## 3. Discussion

This study is the first to put forward accurate PET-derived semi-quantitative thresholds values for GCA aortitis detection using CECT as reference. In addition, the vascular [^18^F]FDG uptake of GCA patients was compared with two different control groups, including an aortic atheroma group, using the same methods. In GCA cases without and with GC before PET/CT, specific aortic cut-off values allowing aortitis to be distinguished from atheromatous lesions, as well as from normal aortic uptake, with a very good sensitivity and specificity have been identified for SUV_max_, TBR _blood_ and TBR_liver_.

Currently, the analysis of aortic [^18^F]FDG uptake is performed visually in comparison to hepatic [^18^F]FDG uptake. This study showed that liver and aortitis SUV were close. Thus, a semi-quantitative analysis, with identification of discriminatory cut-offs values, could be an aid to interpretation and could improve the reproducibility of the analyses. It could be complementary to qualitative analysis and could be useful to automate interpretation; although, the presence of grade 3 [^18^F]FDG uptake of all segments of the aorta and extensive involvement of the supra-aortic trunks and lower limbs are also important diagnostic features of aortitis [10].

Semi-quantitative analysis compared to qualitative analytical approaches could limit the reader bias, increase reproducibility, and optimize precision. Qualitative assessment may be less reliable and accurate compared to semi-quantitative approaches, especially SUV_max_ and TBR_blood_. In addition, SUV_max_ and TBR_blood_ are not yet routinely and easily used PET-based features so our findings are clinically relevant.

Glomerular filtration rate (GFR) can also affect PET-derived features, especially TBR_blood_. Indeed, it has been shown that GFR is negatively associated with FDG distribution in the blood pool [13]. Rosenblum et al. showed that at one-hour imaging the three factors most strongly associated with [^18^F]FDG blood pool background uptake were uptake time, GFR and body mass index [13]. Moreover, in 2014, Besson et al. reported in a semi-quantitative approach to biopsy proven GCA that the aortic to venous blood pool SUV_max_ ratio outperformed the lung and liver ratios [14]. Thus, features involving SUV blood pool could be of interest for aortitis analysis.

In this study, AUC from SUV_max_, TBR_blood_ or TBR_liver_ were high compared to previous studies [15]. This could be explained by the fact that, in this study, every patient had aortitis diagnosed on CT, unlike previous studies which included patients with GCA but without aortic reference imaging (other than PET/CT). However, according to the literature, 50% of GCA patients do not have aortitis at the time of GCA diagnosis, so these previous studies included a number of GCA patients without aortitis, thus lowering the AUC and the SUV_max_.

It is difficult then to determine the best method for PET analysis of large vessel vasculitis in GCA. Qualitative visual assessment requires physician experience and is subjective. Visual PETVAS scoring is more strongly associated with physician interpretation of PET activity rather than TBR or SUV metrics. TBRs outperformed SUV metrics in vascular inflammation in large-vessel vasculitis [11]. The continuous scale of semi-quantitative scoring systems leads to a better ability to discriminate change in PET activity across a wider range of values [11]. Therefore, we propose to mix visual and semi-quantitative analysis in [^18^F]FDG-PET/CT large vessel vasculitis reports.

This study has several limitations, such as its retrospective design, the number of patients included and the maximum interval of 10 days between the start of corticosteroid therapy and [^18^F]FDG-PET/CT imaging. [^18^F]FDG uptake and, consequently, the test’s sensitivity, decreases significantly after GC exposure [18]. In our study, more than one third of GCA patients underwent [^18^F]FDG-PET/CT imaging after the start of steroid therapy. Thus, as in the study by Nielsen et al. [18], we observe a decrease in aortic SUV_max_, TBR_liver_ and TBR_blood_. However, in our study the GCA patients were different as they were included with inflammatory thickenings of the aortic wall on CECT; thus, these structural wall remodelings could favor the persistence of FDG uptake despite more than 3 days of GC treatment. However, in daily practice, the start of GC therapy is often urgent because of the risk of visual impairment, and cannot wait for the PET/CT imaging. Thus, these data seem appropriate for the management of these patients.

Moreover, [^18^F]FDG-PET/CT imaging were performed with analogical devices, thus the results of the semi-quantitative diagnostic cut-off values might be slightly different with a new generation of digital PET/CT devices. Indeed, these new devices offer better spatial resolution reducing the partial volume effect and could improve [^18^F]FDG uptake values. Therefore, the semi-quantitative diagnostic cut-off values found in this work need to be further supported in studies relying on digital PET/CT devices.

The results of this exploratory study need to be confirmed by prospective and multicenter evaluation performed on multiple PET/CT devices.

In conclusion, this study identified discriminative diagnostic cut-off values for different semi-quantitative [^18^F]FDG-PET/CT-derived parameters. These results rely on a CECT reference test which defines aortitis or atheroma. Beyond visual analysis, PET advanced understanding using SUV_max_ and TBR thresholds values could be used for the accurate diagnosis of GCA aortitis. If these results are validated in prospective and multicenter studies, they could be useful for the diagnosis of aortitis with the development of artificial intelligence for the analysis of aortic walls. Thus, SUV_max_ cut-offs of 3.25 and 3.45 could be used to identify GCA aortitis in patients with and without GC before PET.

The use of SUV blood pool values to normalize the interpretation seems interesting since SUV blood is less influenced by corticosteroid therapy than SUV liver. In this way, TBR_blood_, with cut-off values between 1.75 and 1.97 in GCA patients with and without GC before PET, seems to be a good parameter for the analysis of aortic disease with a good AUC, specificity and sensitivity. Multicenter studies are needed to validate these discriminative PET thresholds.

## 4. Materials and Methods

### 4.1. Standard Reference

We used CECT as the reference imaging to analyze the aortic wall. We defined three categories of aortic wall: wall with aortitis, wall with atheroma and normal aortic wall (Figure 3).

This monocentric retrospective study included patients diagnosed with aortitis related to GCA between June 2014 and June 2021. Each GCA case included in this study underwent a CECT and [^18^F]FDG-PET/CT before starting corticosteroid therapy, or within no more than 10 days after its initiation [10].

All GCA patients had to meet at least three American College of Rheumatology (ACR) criteria for the diagnosis of GCA [19], or be over 50 years of age with C-reactive Protein (CRP) ≥ 10 mg/L and large vessel vasculitis.

Aortitis was defined by CECT with a circumferential aortic parietal thickening > 2.2 mm [20] (Figure 3).

Each GCA-related aortitis patient was matched with two aortic atheroma control cases proven on CECT and with one control patient without aortic atheroma (Figure 3). Matching was done on both sex and age. Aortic atheroma control cases had to have at least two out of five CT-positive aortic segments to be included in the study as previously described [10]. The aortic atheromatous patients and control patients were drawn from a group of patients with a history of neoplasia both followed with [^18^F]FDG-PET/CT and CECT.

Aortic atheroma was defined by CECT as an atheromatous lesion with a semi-quantitative ranging ≥1 (score ranging from 0 to 2: 0 for the absence of plaque; 1 for the presence of smooth thin plaques and 2 for the presence of thick irregular plaques (≥3 mm)) [20].

All aortic atheromatous patients and control cases were free of neoplasia at the time of assessment and had not received oncology treatment for at least 3 months. Patients with active cancer or who had been treated within 3 months were excluded.

### 4.2. [^18^F]FDG-PET/CT Acquisition and Analysis

For the PET acquisition method, after at least 6 h of fasting, 3 MBq/kg of [^18^F]FDG was injected intravenously (after recording baseline blood glucose level). After 60 min of resting, [^18^F]FDG-PET/CT imaging was recorded in a supine position from the skull to the base of the thighs with arms next to the body. Images were acquired on a Siemens Biograph mCT64. First, non-contrast CT acquisition was performed with a multi-slice spiral CT scan (Figure 3). Blood glucose levels were measured before [^18^F]FDG injection with a preferred glycemia level ≤ 150 mg/dL; however, up to 200 mg/dL was allowed. Next, a PET acquisition of the same axial range was performed with the patient in the same position. PET data were reconstructed using the Ordinary-Poisson OSEM provided by the manufacturer. All data were corrected for attenuation, scatter and random coincidences. The reconstruction parameters were 3 iterations, 21 subsets and a Gaussian post-filtering of 2 mm FWHM. The voxel size used was 4 × 4 × 2 mm. The time per bed step was adapted following a methodology we previously published [21].

Patients who had focal instead of diffuse [^18^F]FDG uptake in the liver were excluded.

[^18^F]FDG-PET/CT was analyzed using a double blind centralized method; aortic images were segmented according to five anatomical regions: ascending thoracic aorta, aortic arch, descending thoracic aorta, abdominal suprarenal and infrarenal aorta.

An analysis of the different aortic segments was performed by placing Regions of Interest (ROIs) around the vessel in a cross-section. The selected segments were defined according to the Most Diseased Segment (MDS) [22], visually identified, meaning that the slice with the highest standardized uptake value (SUV_max_) was selected, and then the mean of the SUV_max_ from this and the two neighboring slices was calculated.

Different target to background ratios (TBRs) were also recorded by measuring the SUV_max_ of each reference organ. Ratios between aortic wall SUV_max_ and reference site SUV_max_ were evaluated by placing ROIs of similar size (1 cm^3^) (Figure 4):
Target-to-liver ratio (TBR_liver_) by placing a ROI in the healthy right lobe of the liver;Target-to-blood pool ratio (TBR_blood_) defined for supra-diaphragmatic vessels by a ROI drawn centrally in the blood pool of the superior vena cava and for infra-diaphragmatic vessels in the blood pool of the inferior vena cava.

Each of these parameters were compared between the aortitis and control groups and between aortitis and aortic atheroma cases.

Next, an overall aortic analysis was performed by including only the highest values of the five segments per patient. This grouping was done for each PET-based feature. A thoracic aortic analysis was performed by grouping the three thoracic segments, and an abdominal aortic analysis was performed by grouping the two abdominal aortic segments using the same methodology.

### 4.3. Statistical Analysis

Categorical variables were expressed in terms of counts and percentages, and quantitative variables were presented as means ± standard deviations (SD) or medians and inter-quartile range (IQR). The quantitative comparisons were assessed using a student’s t-test or Wilcoxon’s signed rank test in case of variables not normally distributed (assessed by Shapiro–Wilk test). Frequency comparisons were performed using Chi2 or Fisher’s exact test according to the statistical headcount. For all statistical analyses, a two-tailed *p* < 0.05 was considered significant.

For overall thoracic and abdominal aortic analyses, the highest SUV_max_, TBR_blood_ or TBR_liver_ value of the different aortic segments was chosen to perform the analysis.

Area under the ROC curve (AUC) along with a 95% confidence interval (CI) was utilized as a combined measure of sensitivity and specificity to evaluate the performance of each quantitative parameter in PET vs. control or vs. aortic atheroma. AUC values lay between 0 and 1. Metrics with capability to distinguish between binary outcomes will result in an AUC above 0.5, with larger AUC values suggesting better diagnostic performance. The Youden’s J statistic was used to determine the optimal cut-off score that maximized the distance to the identity line. To simplify the visual comparison of ROC curves and diagnostic performances, binomial smoothed ROC curves were produced. ROC curves were performed through pROC R package [23]. R statistical software, version 4.0.4 was used for all statistical analyses.

## Figures and Tables

**Figure 1 ijms-23-15528-f001:**
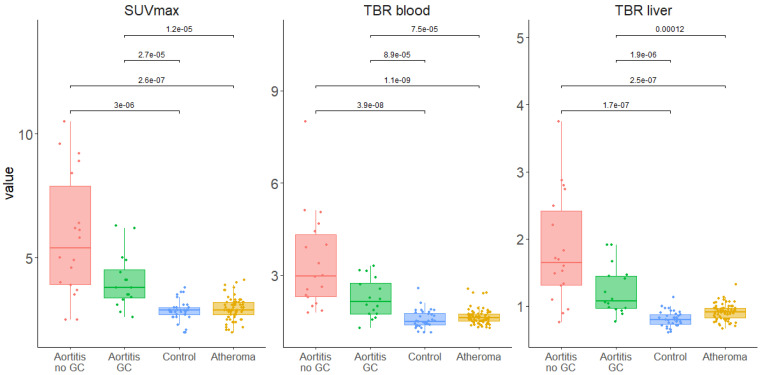
SUV_max_, TBR_blood_ and TBR_liver_ aortic values in an overall aorta analysis in aortitis, aortic atheroma and control groups (GC: glucocorticosteroid).

**Figure 2 ijms-23-15528-f002:**
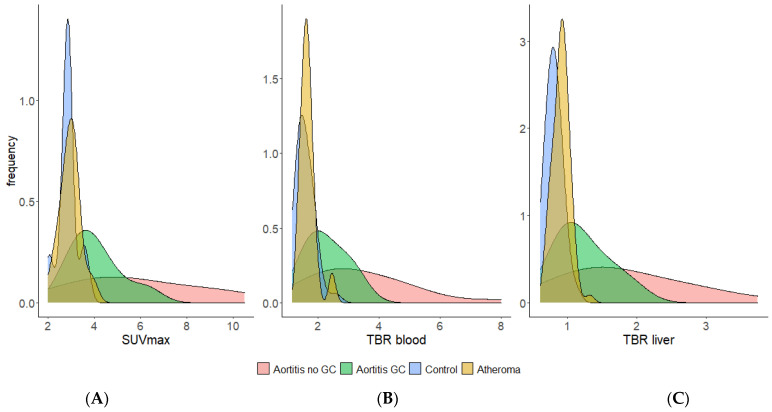
Frequency (density) distribution of SUV_max_ (**A**), TBR_blood_ (**B**) and TBR_liver_ (**C**) values of overall aorta in aortitis, aortic atheroma and control groups.

**Figure 3 ijms-23-15528-f003:**
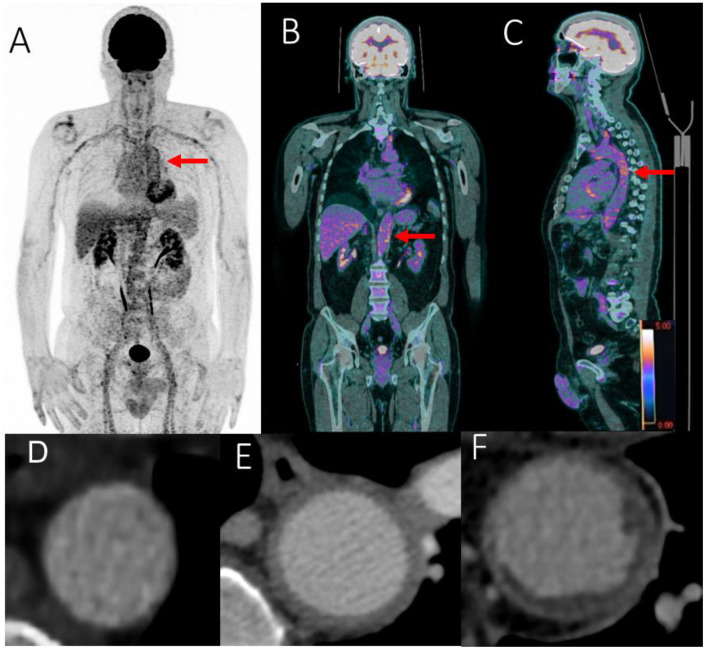
FDG-PET/CT showing grade 3 (red arrows) scattered large vessel vasculitis uptake (including the thoracic and abdominal aorta) in maximum intensity projection (MIP) image (**A**), including the abdominal (**B**) and thoracic aorta (**C**) in coronal and sagittal fused (PET with CT) slices. SUVmax value in the abdominal aortic wall: 6.45. Aortic evaluation with contrast enhanced computed tomography: normal aorta (**D**), aortitis (**E**) and aortic atheroma (**F**).

**Figure 4 ijms-23-15528-f004:**
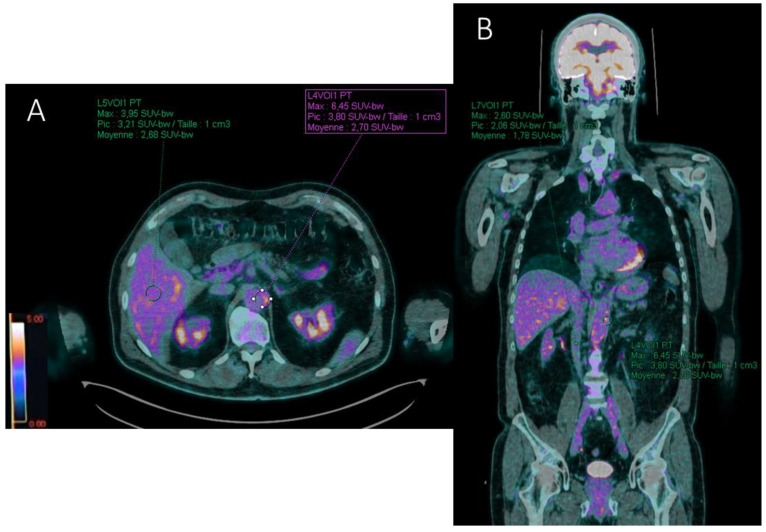
FDG-PET/CT axial (**A**) and coronal (**B**) fused (PET with CT) slices illustrating the computation of the target-to-liver (TBR_liver_) and target-to-blood (TBR_blood_) background ratios in a patient with aortitis. SUV_max_ value of the target (abdominal aortic wall) is recorded with a region of interest (ROI) drawn manually around the arterial structure. Liver (**A**) and blood pool (**B**) background SUV_max_ values are estimated with ROIs projected on the healthy right lobe of the liver and the inferior vena cava, respectively. Here, TBR_liver_ = 6.45/3.95 = 1.63 and TBR_blood_ = 6.45/2.6 = 2.48 for the abdominal aorta.

**Table 1 ijms-23-15528-t001:** Patients’ clinical characteristics (GFR glomerular filtration rate; *p* value 1 GCA aortitis vs. Aortic atheroma; *p* value 2 GCA aortitis vs. control).

	GCA Aortitis *n* = 35	Aortic Atheroma *n* = 70	Control *n* = 35	*p* Value1	*p* Value2
**Median age (Q1–Q3)**	67 (60.7–72.2)	73 (63–77)	70 (61.5–74)	0.21	0.64
**Female sex***n* (%)	30 (85.7)	61 (87.1)	30 (85.7)	0.84	>0.99
**FDG-PET/CT parameters**					
**Mean^18^F-FDG dose** (MBq/kg)	3.0 (±0.06)	3.0 (±0.05)	3.0 (±0.04)	0.09	0.17
**Mean time between injection and imaging** (min) (±SD)	64.8 (±6.8)	63.5 (±7.6)	62.1 (±6.3)	0.38	0.004
**Biological settings**					
**Mean blood glucose level before FDG-PET/CT imaging** (mmol/L) (±SD)	5.3 (±0.98)	5.5 (±1.0)	5.6 (±1.24)	0.34	0.43
**CRP > 10 mg/L***n* (%)	22 (62.9)	8 (11.4)	3 (8.6)	<0.001	<0.001
**CRP** mg/L mean ± SD	65.7 ± 55.2	8.8 ± 30.4	2.7 ± 9.8	<0.001	<0.001
**Glucocorticoids before PET***n* (%)	16 (45.7)	9 (12.6)	2 (5.7)	0.001	<0.001
**Patients with hepatic cytolysis***n* (%)	1 (2.9)	14 (20.0)	6 (17.1)	0.02	0.04
**GFR** mL/min(±SD)	81.5 (±16.6)	80.2 (±20.0)	85.8 (±17.8)	0.70	0.30
**GFR <60 mL/min***n* (%)	4 (11.4)	8 (11.4)	3 (8.5)	>0.99	0.69

**Table 2 ijms-23-15528-t002:** SUV_max_ liver, blood and highest aortic SUV_max_ according to aortitis, aortic atheroma and control groups (GCA: giant cell arteritis; GC: glucocorticosteroid).

Mean; Median [IQR]	GCA Aortitis without GC *n* = 19	GCA Aortitis Treated with GC *n* = 16	Aortic Atheroma *n* = 70	Control *n* = 35	*p*
**SUV_max_ liver**	3.3;3.2 [2.95, 3.40]	3.4;3.4 [3.08, 3.52]	3.3;3.2 [2.95, 3.60]	3.6;3.4 [3.20, 3.90]	0.047
**SUV_max_ blood**	1.8;1.9 [1.50, 1.95]	1.8;1.9 [1.58, 2.02]	1.8;1.8 [1.60, 2.00]	1.9;1.8 [1.60, 2.10]	0.697
**SUV_max_ aortic**	5.9;5.5 [4.25, 7.40]	4.1;3.9 [3.38, 4.53]	2.9;2.90 [2.65, 3.20]	2.9;2.9 [2.70, 3.00]	<0.001
**TBR liver**	1.8;1.6 [1.31, 2.35]	1.2;1.1 [0.97, 1.45]	0.8;0.8 [0.73, 0.88]	0.9;0.9 [0.82, 0.97]	<0.001
**TBR blood**	3.4;3.0 [2.33, 4.21]	2.3;2.1 [1.84, 2.76]	1.6;1.5 [1.39, 1.77]	1.7;1.6 [1.49, 1.81]	<0.001

**Table 3 ijms-23-15528-t003:** Semi-quantitative aortic FDG uptake according to the visual grade of FDG uptake.

Aortic Visual Grading	SUV_max_	TBR_liver_	TBR_blood_
Grade 0	2.6 ± 0.4	0.8 ± 0.1	1.4 ± 0.3
Grade 1	2.8 ± 0.5	0.9 ± 0.1	1.6 ± 0.3
Grade 2	3.6 ± 0.5	1.0 ± 0.2	1.9 ± 0.4
Grade 3	5.1 ± 1.7	1.6 ± 0.5	2.9 ± 1.2

**Table 4 ijms-23-15528-t004:** Aortic PET values in aortitis with glucocorticosteroid vs. all control patients (aortic atheroma patients and normal aortic control patients) receiver operating characteristic (ROC) curve analyses, and aortitis without glucocorticosteroid vs. all control patients ROC curve analyses in thoracic and abdominal aorta and in overall aorta (AUC: area under the curve, GC: glucocorticoids, CI: confidence interval).

	Aortitis with GC vs. All Controls			Aortitis without GC vs. All Controls		
	SUV_max_	TBR Blood	TBR Liver	SUV_max_	TBR Blood	TBR Liver
**Overall aorta**						
AUC [CI 95%]	0.86 [0.74;0.98]	0.83 [0.69;0.96]	0.85 [0.73;0.96]	0.89 [0.76;1]	0.96 [0.93;1]	0.91 [0.81;1]
Cut-off	3.25	1.75	0.97	3.45	1.97	1.09
Specificity	0.83	0.75	0.82	0.90	0.93	0.96
Sensitivity	0.81	0.81	0.75	0.89	0.89	0.83
**Thoracic aorta**						
AUC [CI 95%]	0.89 [0.79;0.99]	0.84 [0.73;0.96]	0.89 [0.79;0.98]	0.90 [0.78;1]	0.96 [0.93;1]	0.93 [0.84;1]
Cut-off	3.25	1.75	0.97	3.45	1.77	1.09
Specificity	0.81	0.80	0.80	0.93	0.81	0.97
Sensitivity	0.88	0.81	0.88	0.88	1	0.82
**Abdominal aorta**						
AUC [CI 95%]	0.89 [0.79;0.98]	0.93 [0.86;1]	0.89 [0.80;0.98]	0.90 [0.78;1]	0.94 [0.86;1]	0.94 [0.85;1]
Cut-off	3.05	1.5	0.91	3.95	1.81	1.13
Specificity	0.76	0.71	0.73	0.99	0.96	1
Sensitivity	0.91	1	0.91	0.75	0.88	0.88

## Data Availability

The data is available upon request from the corresponding author.

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
