# Peer review of "Semi-Quantitative [18F]FDG-PET/CT ROC-Analysis-Based Cut-Offs for Aortitis Definition in Giant Cell Arteritis"

_ijms, 2022, doi:10.3390/ijms232415528_

Round 1
Reviewer 1 Report
General comment : this is an interesting paper on the role of quantitative and semi-quantitative indices to discriminate LVV vs atheroma and controls. The paper is well written but suffers a number of flaws. In particular, there is no group (or it is not defined in the manuscript) of patients referred for GCA/LVV and eventually with a negative diagnosis, which is a bias.
Specific comments:
1. From the ROC curve analysis, the AUC are exceptionally high which is hard to believe, and this is due to the above inclusion criterion; the role of a test should be to discriminate between patients with a suspicion that eventually turn out to be negative, after the whole workup. There is no mention of results or in the discussion about this.
2. The use of SUVmax aorta to SUVmean liver or blood is not well described in Mat/Meth and is unclear in the Figure 1. When looking at the heterogeneity of the liver uptake as shown in Figure 4, the SUVmean of the liver be much higher if a larger ROI was drawn on the right lobe. My feeling is that the cutoff could vary a lot depending on the selection of the liver ROI.
3. The authors included patients treated with glucocorticoids for up to 10 days and show that the diagnostic value is not far to those untreated: this is in total contradiction with the data of Nielsen et al. who showed that three days were ok, but not 10 days (if I read correctly, this paper is not cited). But the data are not clearly presented in a tabular format but only in Figure 2, as histograms.
4. On lines 202-204, the authors claim (in bold) that semiquantitative indices (TBR thresholds) could be used in difficult cases! If the indices are robust, they could be used in ALL cases as a sort of artificial intelligence. It a method is only usable in difficult case, it is definitely not robust, independent.
5. The study ran between 2014 and 2021: it could have been an exploratory study, followed by a prospective confirmatory study. The authors could elaborate on this.
Minor comments:
1. In Table 1, p1 and p2 are confusing: they should better read as P value 1 and P value 2
2. In Table 1, the actual values of CRP should be given as mean +/- s.d. rather than only those with the 10 mg/L ACR criterion
3. In Table 1, I do not see the relevance of liver enzyme disturbance: is it meant to make any correction to the liver SUVmean?
4. In table 2, as the authors present the IQR, they should omit either the(mean or) median: this makes the Table hard to read.
5. Axis legends in Fig. 2 are two small, unreadable.
Author Response
Dear reviewer,
Thank you for your comments. We have provided a response to each question raised.
The detailed answers and the revised manuscript are attached.
Sincerely

Reviewer 2 Report
I read the manuscript. Here are the comments from this reviewer.
Abstract
1. It states that FDG-PET was performed in 35 GCA cases. As noted by the authors, approximately 50% of GCA patients show aortitis. It is difficult to read whether the analysis included patients with and without aortitis, or whether the analysis was only for GCA patients with aortitis.
2. I think that the abstract should also indicate how many untreated and treated cases were included.
3. Expressions such as glucocorticoids, corticosteroids, and steroids are different, so please unify them.
Results
1. In this study, the cutoff between normal control and GCA aortitis, or between atheroma and GCA aortitis, was calculated. But at the end of the results, can you compare GCA aortitis vs normal control and atheroma combined and give a cut-off value, because in clinical practice, it is important whether it is aortitis or not?
2. The figure label is too small to read.
3. Since there are two Table 3, correction is necessary.
Author Response
Dear reviewer,
Thank you for your comments. We have provided a response for each question raised.
The detailed answers and the revised manuscriptare attached .
Sincerely

Round 2
Reviewer 1 Report
Thanks to the authors for addressing most issues. My feeling is that the paper has become more complicated. Looking at the abstract, the results are perfect, but do not fit with the results section.
What the potential readers needs is actually what to do in patients with aortitis and a suspicion of GCA. This should be straightforward from the abstract and not to be looked at in complicated results section.
The authors should try to summarize the ROC AUC (because it is the basis of their conclusion) in the abstract and later to focus on the main findings of these AUCs (putting back the raw data in a supplemental file). If I read correctly, the authors present the AUC of the ROC curves without variance or confidence intervals: what we actually need is AUC+variance and P values according to the 90% confidence interval and this is missing.
Thanks to the authors to have split the analysis of patients on and not already on glucosteroids.
There is about this a small mistake (inversion) between lines 24 (54% off) and line 94 (54% on) but this has probably little impact on the MS.
More generally, this paper deserves to be published, but In a way audience can understand it. It the current format, it is too complicated
Author Response
Response to Reviewer 1 Comments:
Thanks to the authors for addressing most issues. My feeling is that the paper has become more complicated. Looking at the abstract, the results are perfect, but do not fit with the results section.
What the potential readers needs is actually what to do in patients with aortitis and a suspicion of GCA. This should be straightforward from the abstract and not to be looked at in complicated results section.
The authors should try to summarize the ROC AUC (because it is the basis of their conclusion) in the abstract and later to focus on the main findings of these AUCs (putting back the raw data in a supplemental file). If I read correctly, the authors present the AUC of the ROC curves without variance or confidence intervals: what we actually need is AUC+variance and P values according to the 90% confidence interval and this is missing.
We agree with the reviewer, we have simplified the manuscript and left only the table assessing aortic disease versus control with and without glucocorticoids in the results others results are now in supplemental.
We have summarized the AUC ROCs in the abstract. For each AUC presented in Table 4 we have added the 95% confidence intervals
Thanks to the authors to have split the analysis of patients on and not already on glucosteroids.
There is about this a small mistake (inversion) between lines 24 (54% off) and line 94 (54% on) but this has probably little impact on the MS.
We have corrected the version in Table 2, On line 94, GCA patients are glucocorticosteroid-free before PET.
Reviewer 2 Report
all questions raised by this reviewer were addressed and the revised manuscript is now acceptable to this journal.
Author Response
Thank you for your comments.
Round 3
Reviewer 1 Report
+Thanks for making it simpler. Accepted.